



# Global Spatial Variation in the PM$_{2.5}$ to AOD Relationship Strongly
# Influenced by Aerosol Composition
Haihui Zhu[1]*, Randall V. Martin[1], Aaron van Donkelaar[1], Melanie S. Hammer[1], Chi Li[1], Jun Meng[2],
Christopher R. Oxford[1], Xuan Liu[1], Yanshun Li[1], Dandan Zhang[1], Inderjeet Singh[1], Alexei Lyapustin[3]
[1]Department of Energy, Environmental & Chemical Engineering, Washington University in St. Louis, St.
Louis, MO, USA
[2]Department of Civil and Environmental Engineering, Washington State University, Pullman, WA, USA
[3]Laboratory for Atmospheres, NASA Goddard Space Flight Center, Greenbelt, MD, USA
*Correspondence*: Haihui Zhu (haihuizhu@wustl.edu)
**Abstract** Ambient fine particulate matter (PM$_{2.5}$) is the leading global environmental determinant
of mortality. However, large gaps exist in ground-based PM$_{2.5}$ monitoring. Satellite remote sensing
of aerosol optical depth (AOD) offers information to fill these gaps worldwide, when augmented
with a modeled PM$_{2.5}$ to AOD relationship ($\eta$). This study aims to understand the spatial pattern
and driving factors of $\eta$ from both observations and modeling. A global observational estimate of
$\eta$ for the year 2019 is inferred from 6,118 ground-based PM$_{2.5}$ measurement sites and satellite
retrieved AOD from the MAIAC algorithm. A global chemical transport model, GEOS-Chem, in
its high performance configuration (GCHP), is used to interpret the observed spatial pattern of
annual mean $\eta$. Measurements and the GCHP simulation consistently identify a global population-
weighted mean $\eta$ of 92 – 100 μg/m$^3$, with regional values ranging from 60.3 μg/m$^3$ for North
America to more than 130 μg/m$^3$ in Africa. The highest $\eta$ is found in arid regions where aerosols
are less hygroscopic due to mineral dust, followed by regions strongly influenced by surface
aerosol sources. Relatively low $\eta$ is found over regions distant from strong aerosol sources. The
spatial variation of $\eta$ is strongly influenced by aerosol composition driven by its effects on aerosol
hygroscopicity. Sensitivity tests with globally uniform parameters reveal that aerosol composition
leads to the strongest $\eta$ spatial variability, with a population-weighted normalized mean difference
of 12.3 μg/m$^3$, higher than that from aerosol vertical profile (8.4 μg/m$^3$), reflecting the determinant
composition effects on aerosol hygroscopicity and aerosol optical properties.



**1 Introduction**
Exposure to ambient fine particulate matter ($PM_{2.5}$) has been recognized as the predominant
environmental risk factor for the global burden of disease, leading to millions of deaths annually
(Murray et al., 2020; Burnett et al., 2018). Even at low $PM_{2.5}$ concentrations, long-term exposure
can increase circulatory and respiratory related mortality (Pinault et al., 2016; Christidis et al.,
2019; Weichenthal et al., 2022). Despite the importance of $PM_{2.5}$, many of the world's countries
do not provide publicly accessible $PM_{2.5}$ data (Martin et al., 2019). Satellite remote sensing of
aerosol optical depth (AOD), an optical measure of aerosol abundance, offers information about
the distribution of $PM_{2.5}$ (Kondragunta et al., 2022). A large community relies upon the spatial
distribution of $PM_{2.5}$ concentrations inferred from satellite remote sensing for health impact
assessment and epidemiological analyses of long-term exposure (Murray et al., 2020; Burnett et
al., 2018; Hao et al., 2023; Cohen et al., 2017). Quantitative application of satellite AOD for long-
term characterization of the spatial distribution of $PM_{2.5}$ would benefit from a better understanding
of the factors affecting the $PM_{2.5}$-AOD relationship.
The relationship between satellite AOD and surface $PM_{2.5}$ can be established through a statistical
method, a geophysical method, or their combination. A statistical method uses ground-based
monitors for training and is well suited for regions with dense monitors (Xin et al., 2014; Hu et al.,
2014; Di et al., 2016). A geophysical approach utilizes a chemical transport model to simulate the
relationship (η) between $PM_{2.5}$ and AOD for application to satellite AOD (van Donkelaar et al.,
2006, 2010; He et al., 2021), and thus depends on accurate model representation of η. van
Donkelaar et al. (2015, 2016) combined the two methods by applying geographically weighted
regression (GWR) on the geophysical $PM_{2.5}$, which further constrains geophysical $PM_{2.5}$ using
ground measurements and other predictors. However, accuracy of geophysical $PM_{2.5}$ remains
critical over vast areas with sparse monitoring, and knowledge about the factors affecting η spatial
variability are needed to guide improvements of modeled η and geophysical $PM_{2.5}$.
Previous studies have identified several factors that affect η variability, including aerosol vertical
distribution, aerosol hygroscopicity, aerosol optical properties, and ambient meteorological factors
such as relative humidity (RH), planetary boundary layer height (PBLH), wind speed, temperature,
and fire events (Wendt et al., 2023; Jin et al., 2019; Guo et al., 2017; Ford and Heald, 2015; Li et
al., 2015; van Donkelaar et al., 2013). Most studies focused on the temporal variability of η and



found association with meteorological variables such as PBLH (Yang et al., 2019; He et al., 2021;
Chu et al., 2015; Gupta et al., 2006; Zhang et al., 2009; Damascena et al., 2021). A few studies
have examined the regional-scale spatial variation of η with meteorological, land type variables,
and aerosol vertical profile in North America (Jin et al., 2020; van Donkelaar et al., 2006; Li et al.,
2015) and China (Yang et al., 2019). To our knowledge, none have examined the factors at the
global scale affecting the spatial variation of η or the effects of chemical composition.
In this work, we examine this knowledge gap about the spatial variation in η at a global scale. We
first collect data from more than 6,000 $PM_{2.5}$ monitoring sites provided by nine networks and
satellite AOD to obtain an observationally based map of η. We further interpret the global η using
the GEOS-Chem model of atmospheric composition with recent improvements in aerosol size
representation, $PM_{2.5}$ diel variation, and vertical allocation. By decomposing the simulated η, we
identify 2 strong drivers of η spatial variability: aerosol composition and aerosol vertical profile.
We conduct sensitivity tests using GEOS-Chem to study how the two factors vary globally and
how they contribute to the spatial variation in η.

## 72  2   Methods

### 73  2.1   Ground Measured $PM_{2.5}$

We collect ground-based measurements of $PM_{2.5}$ for the year 2019 from which to produce
observational constraints on η. We obtain $PM_{2.5}$ measurements from 7 regional networks and 2
global networks, as shown in Figure A1. For the United States, we access data from the United
States Environmental Protection Agency's Air Quality System (https://www.epa.gov/outdoor-air-
quality-data/download-daily-data), including both Federal Reference Method and non-Federal
Reference Methods $PM_{2.5}$ (e.g. IMPROVE network). $PM_{2.5}$ data for Canada are from the
Environment Canada's National Air Pollution Surveillance (NAPS) program. $PM_{2.5}$ data for
Europe are from the European Environment Agency Air Quality e-Reporting system
(https://www.eea.europa.eu/data-and-maps/data/aqereporting). Over mainland China, $PM_{2.5}$
measurements are downloaded from http://beijingair.sinaapp.com/, which provides instantaneous
air quality data records from the National and Provincial Environmental Protection Agencies. Over
India, $PM_{2.5}$ data are originally from the Central Pollution Control Board Continuous Ambient Air
Quality Monitoring network and the U.S. embassies. Over Australia, observations are downloaded





for    the    Northern    Territory    (http://ntepa.webhop.net/NTEPA/),    Queensland
(https://www.data.qld.gov.au/dataset/), and New South Wales (https://www.dpie. nsw.gov.au/air-
quality/air-quality-data-services/data-download-facility).    We    require    at    least    5    days    of
measurements for each month for a monitor to be included. Additionally, we obtain $PM_{2.5}$
measurements over other regions provided by the World Health Organization (WHO) Global
Ambient Air Quality Database and by the Surface PARTiculate mAtter Network Network
(SPARTAN), which is co-located with the Aerosol Robotic Network (AERONET). SPARTAN
also provides filter based $PM_{2.5}$ chemical composition, which is initially described in Snider et al.,
(2016). Subsequent developments to the sampling and analysis procedure of SPARTAN include
an upgrade to the AirPhoton SS5 sampling station to use a cyclone inlet, an automated weighing
system (MTL AH500E) to improve precision and throughput, additional black carbon analysis by
Hybrid Integrating Plate/Sphere (White et al., 2016), trace metal elements measured by X-ray
Fluorescence (Liu et al., 2024) and a global mineral dust equation (Liu et al., 2022). We require
50 days of coincident $PM_{2.5}$ and AERONET AOD measurements for a SPARTAN site to be
included in our analysis.

## 2.2    Satellite AOD

We obtain AOD at 550 nm from the Multi-Angle Implementation of Atmospheric Correction
(MAIAC) algorithm, which offers AOD at a high spatial resolution of 1 km worldwide over both
land and coastal regions (Lyapustin et al., 2018). The radiances used in the retrieval are measured
by the twin MODerate resolution Imaging Spectroradiometer (MODIS) instruments onboard the
Terra and Aqua satellites. Terra follows a descending orbital path, crossing the equator at 10:30
local time, while Aqua is on an ascending orbit with 13:30 equatorial crossing local time. Both
MODIS instruments offer a wide swath width of 2330 km, enabling nearly global daily coverage
of the Earth (Sayer et al., 2014). $PM_{2.5}$ monitoring sites with annual mean satellite AOD less than
0.05 (background AOD level over land) are excluded to reduce the influence of retrieval
uncertainties on our analysis.

## 2.3    AERONET AOD

AERONET is a worldwide sun photometer network that provides long-term measurement of AOD.
We use the Version 3 Level 2 database, which includes an improved cloud screening algorithm





(Giles et al., 2019). We sample AERONET AOD within ±15 min of the satellite overpass time and
interpolate to 550 nm wavelength, based on the local Ångström exponent at 440 and 670 nm. For
SPARTAN sites, we sample AERONET data coincidentally with SPARTAN aerosol composition
to obtain the ground-based observation of η.

## 120    2.4    GEOS-Chem Simulation

We simulate η with the GEOS-Chem chemical transport model (www.geos-chem.org, last access:
26 October 2023), driven by offline meteorological data, MERRA-2, from the Goddard Earth
Observing System (GEOS) of the NASA Global Modeling and Assimilation Office (Schubert et
al., 1993). We use the high-performance configuration of GEOS-Chem (GCHP) (Eastham et al.,
2018) version 13.4.0 (DOI: 10.5281/zenodo.7254268), which includes advances in performance
and usability (Martin et al., 2022). The simulation is conducted for the year 2019, on a C90 cubed-
sphere grid corresponding to a horizontal resolution of about 100 km, with a spin-up time of 1
month.
The GEOS-Chem aerosol simulation includes the sulfate-nitrate-ammonium (SNA) system
(Fountoukis and Nenes, 2007), primary and secondary carbonaceous aerosols (Park et al., 2003;
Wang et al., 2014; Pai et al., 2020), sea salt (Jaeglé et al., 2011), and natural (Fairlie et al., 2007;
Meng et al., 2021) and anthropogenic (Philip et al., 2017) dust. The primary emission data are
from the Community Emissions Data System (CEDS$_{GBD-MAPS}$; McDuffie et al., 2020). Emissions
from stacks are distributed vertically (Bieser et al., 2011). Diel variation of anthropogenic
emissions is included (Li et al., 2023). Resolution-dependent soil $NO_x$, sea salt, biogenic VOC,
and natural dust emissions are calculated offline at native meteorological resolution to produce
consistent emissions across resolutions (Weng et al., 2020; Meng et al., 2021). Biomass burning
emissions use the Global Fire Emissions Database, version 4 (GFED4) at daily resolution (van der
Werf et al., 2017). We estimate organic matter (OM) from primary organic carbon using an
OM/OC parameterization (Philip et al., 2014; Canagaratna et al., 2015). For secondary aerosol
components, the concentration at 2 m above the surface is used to calculate PM$_{2.5}$, following Li et
al. (2023). A 50% reduction of the surface nitrate concentration is applied to account for the long-
persisting bias in surface nitrate simulated by GEOS-Chem (Heald et al., 2012; Zhang et al., 2012;
Zhai et al., 2021; Miao et al., 2020; Travis et al., 2022) and other models (Zakoura and Pandis,



2018; Shimadera et al., 2014). Dry and wet deposition follows Amos et al. (2012), with a standard
resistance-in-series dry deposition scheme (Wang et al., 1998). Wet deposition includes
scavenging processes from convection and large-scale precipitation (Liu et al., 2001).
Global RH-dependent aerosol optical properties are based on the Global Aerosol Data Set (GADS)
(Kopke P., 1997), as originally implemented by Martin et al. (2003), with updates for SNA and
OM dry size (Zhu et al., 2023), hygroscopicity (Latimer and Martin, 2019), mineral dust size
distribution (Zhang et al., 2013), and absorbing brown carbon (Hammer et al., 2016). We
artificially increase simulated AOD by 0.04 globally to address a poorly understood systematic
bias. $PM_{2.5}$ is calculated as the sum of each component at 35% RH to align with common
measurement protocols.
**2.5   Population**
Global population information is obtained from the Gridded Population of the World provided by
the NASA Socioeconomic Data and Applications Center (Center for International Earth Science
Information Network - CIESIN - Columbia University, 2018).
**2.6   Sensitivity Tests with Globally Uniform Parameters**
We conduct sensitivity tests of factors affecting the spatial variation of η, with a focus on aerosol
composition and aerosol vertical profile. To understand the relative importance of these factors,
we impose a constant for each factor and simulate the corresponding η. The difference between
the test scenario and the base scenario reflects the change due to variation of the factor. We use
the global population-weighted mean (PWM) and population-weighted mean difference (PWMD)
to summarize changes with a focus on relevance to population exposure:

$$X_{PWM} = \frac{\sum_j \sum_i P_{i,j} X_{i,j}}{\sum_j \sum_i P_{i,j}}$$


$$PWMD = \frac{\sum_j \sum_i P_{i,j} |X_{i,j} - Y_{i,j}|}{\sum_j \sum_i P_{i,j}}$$

where i and j are grid box identifiers. X and Y could be any variable of interest. $|X_{i,j} - Y_{i,j}|$ is the
absolute value of their difference.



The first test imposes globally uniform aerosol chemical composition calculated as the global
PWM aerosol component fraction ($F_{s,k,PWM}$):

$$F_{s,k,PWM} = \frac{\sum_j \sum_i P_{i,j} F_{i,j,s,k}}{\sum_j \sum_i P_{i,j}}$$

where i, j, and k are grid box identifiers along latitude, longitude, and vertical layer. P represents
population density in each grid box. $F_S$ is the fraction of aerosol component S in total aerosol mass.
This test keeps the total columnar aerosol mass and aerosol vertical profile unchanged.
The second test imposes a globally uniform aerosol vertical profile calculated as the PWM column
relative vertical profile ($R_{s,k,PWM}$):

$$R_{s,k,PWM} = \frac{\sum_j \sum_i P_{i,j} R_{i,j,k,s}}{\sum_j \sum_i P_{i,j}}$$

where $R_{i,j,k,s}$ is the relative dry mass ratio compared to the surface. The total mass loading and
relative chemical composition are unchanged.
We analyze global and regional variations of η, as well as that for the driving factors. The definition
of each region used in this study is summarized in Figure A2.

## 3  Results and Discussion

### 3.1  Global Spatial Pattern of η

The top panel of Figure 1 shows the observationally based annual mean η, inferred from the ratio
of ground-measured $PM_{2.5}$ to MAIAC AOD. Measurements are most dense in North America,
Europe, and East Asia. The annual mean η varies substantially, from 7.8 μg/m³ in Hawaii to 321
μg/m³ in Central Asia, with a PWM of 95.7 μg/m³ and standard deviation (σ) of 36.6 μg/m³. Higher
PWM η of 132 μg/m³ to 154 μg/m³ exist over desert regions such as the North Africa and West
Asia, followed by PWM η of 97 μg/m³ to 121 μg/m³ by regions strongly influenced by
anthropogenic aerosols, such as East Asia, South Asia (Figure A3 and Table A1). Over North
America, η is around 60 μg/m³ in the east and in California, which is more than double that in the
Rockies, driven by the spatial pattern of surface $PM_{2.5}$ (Figure A3). The PWM η in North America
of 60.3 μg/m³ is about 30% lower than the global PWM. The η pattern found here is similar to that
reported by Jin et al. (2020) for the U.S. In Europe, η also varies noticeably between the east and



the west, driven by the spatial pattern of surface $PM_{2.5}$, as $PM_{2.5}$ increases by 60% from west to
east while AOD increases by only 8%. The PWM η in Europe is 94.0 μg/m³, slightly lower than
the global PWM. In Asia, measured η is concentrated in China and India. In China, the η spatial
pattern shows a clear distinction between the northern and southern regions, driven by the higher
AOD in the south, where relative humidity is high. A similar η spatial pattern and a negative
correlation between η and RH are reported by Yang et al. (2019). In India, η is highest in the
northwest, with a PWM η of 128 μg/m³, and decreases to about 80 μg/m³ toward the east and the
south. Both $PM_{2.5}$ and AOD follow the same spatial pattern, while $PM_{2.5}$ exhibits a stronger
decreasing tendency. PWM η in Asia is 100 μg/m³, the highest among the populous regions and
4.5% higher than the global PWM. Globally, from west to east, η increases by about 62%, despite
that both $PM_{2.5}$ and AOD increased more than threefold (Figure A5). The coefficient of variation
(standard deviation divided by mean) in η is higher in Europe (μ = 0.31) and Asia (μ = 0.34), than
North America (μ = 0.23, Figure A5).
The bottom panel in Figure 1 shows the GCHP simulated η, the ratio between simulated 24-hour
mean surface $PM_{2.5}$ and simulated total column AOD at satellite overpass time. The simulation
generally reproduces the global observations of η with a tendency for high values in arid regions
influenced by dust and low values in regions distant from strong surface sources. The global
simulated PWM η is 4% higher than the observations (99.5 μg/m³ vs. 95.7 μg/m³), mostly driven
by an overestimation in Asia (107 μg/m³ vs. 100 μg/m³), that reflects an overestimation of PWM
$PM_{2.5}$ (47.0 μg/m³ vs. 43.6 μg/m³). The simulation generally reproduces the regional spatial pattern
in North America and Asia but underestimates the η variability in Europe as it overestimates η in
central Europe and underestimates η in Eastern Europe, due to positive bias in simulated $PM_{2.5}$ in
central Europe and positive bias in simulated AOD in Eastern Europe. Nonetheless, the PWM η
in Europe (84.1 μg/m³) is within 11% of observations. Globally, there is overall consistency
between the simulated η and observed η, with a correlation of 0.64, resulting in a high degree of
consistency between geophysical $PM_{2.5}$ and measured $PM_{2.5}$ (r = 0.90, Figure A5).



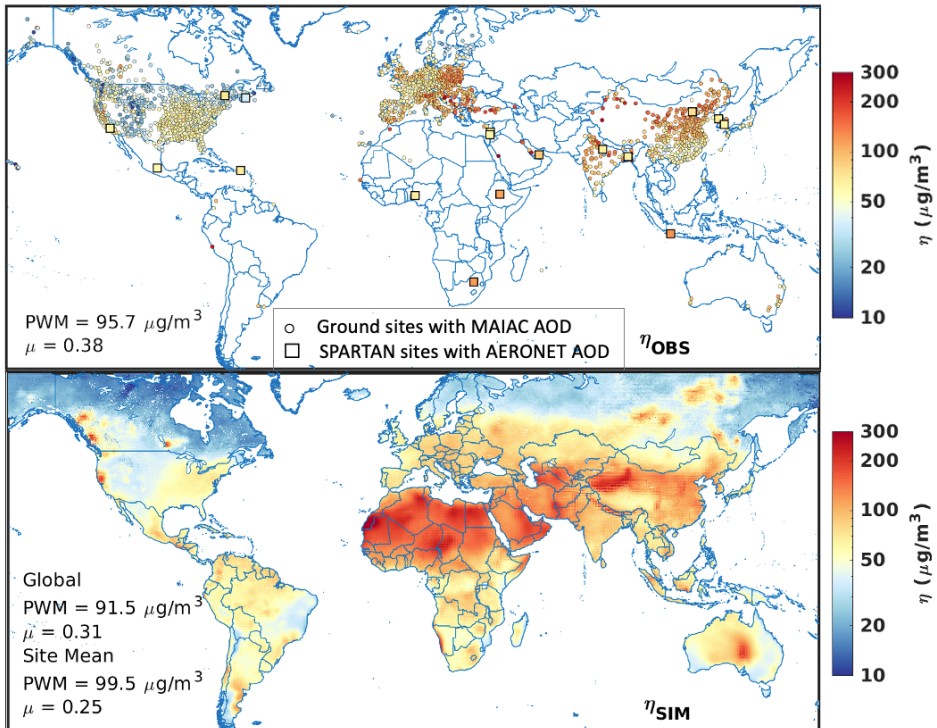


Figure 1. Observed (top) and simulated (bottom) annual mean η for 2019. Circles represent ground
measurement sites from regional networks or the World Health Organization. Squares represent co-
located ground measured PM$_{2.5}$ from SPARTAN and AOD from AERONET. PWM = population-
weighted mean, μ = coefficient of variation (standard deviation divided by mean).
We explore the dominant driving factors for η spatial variation by calculating the spatial
correlation between each candidate factor and the observation-based η. Candidate factors
examined include meteorological fields (MERRA-2), aerosol vertical profile, and aerosol
composition as collected from the GCHP simulation or SPARTAN. Meteorological fields include
those commonly considered to represent the temporal variation in η, such as PBLH, RH at 700
hPa, wind speed at 10 m, and temperature at 2 m (Yang et al., 2019; He et al., 2021; Chu et al.,
2015; Damascena et al., 2021). The aerosol vertical profile is represented as the AOD fraction
below 1 km (AOD % below 1 km). Aerosol composition includes SNA, OM, dust, black carbon,
and sea salt, all represented as the fractional contributions (%) to surface PM$_{2.5}$. Figure 2 shows
the spatial correlation of annual mean factors versus observation-based η. Aerosol components,





particularly those with strong primary sources (dust, OM, and black carbon), exhibit the strongest
correlations (>0.3) with observationally based η. Significant positive correlations are found for
mineral dust and black carbon, both of which are non- or weakly-hygroscopic. Significant negative
correlations are found for organic matter and sea salt, reflecting a weak connection between surface
concentrations and AOD aloft. Processes are further discussed in sections 3.2 and 3.4. The aerosol
vertical profile exhibits a moderate correlation with η (0.18), which is notably higher than any
meteorological factors (<0.12). Ground-based data from SPARTAN and AERONET corroborate
the correlation between aerosol composition and η (Figure A6).
The indicators of spatial variation in η found here differ from that for temporal variation of η in
prior work (e.g. He et al. 2021), reflecting the different processes involved. Meteorological
parameters drive short-term variability in the aerosol vertical profile, such as day-to-day variation
in mixed layer depth or in advection from a point source. In contrast, the spatial variation in annual
mean η reflects the spatial variation in processes affecting the long-term relation of surface $PM_{2.5}$
at controlled RH of 35% with AOD at ambient RH. Aerosol composition and the aerosol vertical
profile reflect spatial variation in aerosol hygroscopicity, mass extinction efficiency, and sources.
The following sections explore how aerosol composition and aerosol vertical profile vary globally
and examine how they affect the spatial pattern of η by conducting two sensitivity tests. In each
sensitivity test, we replace the spatial variability of a factor with a globally uniform value. The
variability of aerosol composition and aerosol vertical profile are discussed in sections 3.2 and 3.3,
respectively. The sensitivity test results are discussed in section 3.4.



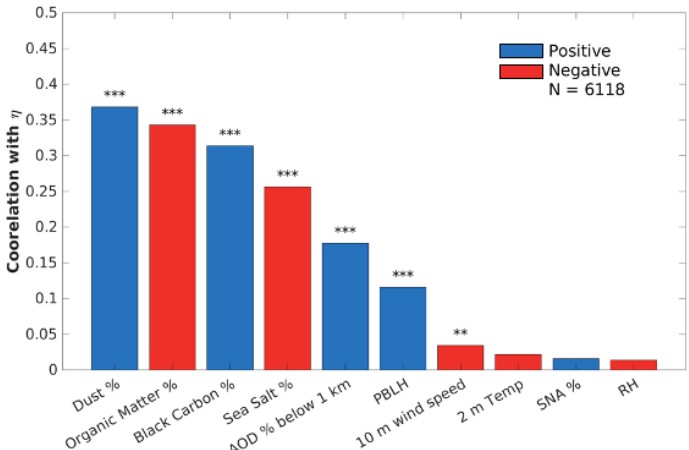


Figure 2. Spatial correlation between annual mean modeled parameters and observationally-based η. Blue
bars indicate positive correlations. Red bars indicate negative correlations. Stars above each bar indicate
the p-value associated with each correlation. '***' indicates the p-value is lower than 0.001 and '**'
indicates lower than 0.01.

## 3.2 Spatial Variability in Aerosol Composition

Figure 3 shows the simulated PWM aerosol composition globally and regionally, as well as the
global area-weighted mean (AWM). The top panel shows the compositional contribution to $PM_{2.5}$.
Globally, dust is the leading PWM $PM_{2.5}$ component (34.7%), followed by OM (31.9%) and SNA
(29.3%). The bottom panel shows the compositional contribution to AOD. PWM AOD
composition is more evenly distributed, with more contribution from SNA (49.9%), followed by
OM (27.2%) and dust (16.1%). Overall, more hygroscopic aerosols such as SNA tend to contribute
a larger fraction of AOD which is at ambient RH, while less hygroscopic aerosols, such as mineral
dust tend to contribute a larger fraction of $PM_{2.5}$ which is at controlled RH of 35%. The AWM
$PM_{2.5}$ and AOD composition exhibit weaker contributions from SNA, primarily reflecting a larger
contribution from dust in remote regions than in more densely populated areas. Over populous
regions such as North America, Europe, and Southeast Asia, there are greater SNA and OM
fractions than the global mean (Figure 3). Arid regions, such as West Asia, the Middle East, North
Africa, and Sub-Saharan Africa, have large fractions of non-hygroscopic mineral dust that (1)
reduce aerosol mass extinction efficiency, yielding less AOD per unit mass, and (2) are unaffected



by the controlled RH of $PM_{2.5}$. Both of these factors increase η in dusty regions compared with
regions dominated by hygroscopic SNA aerosols.

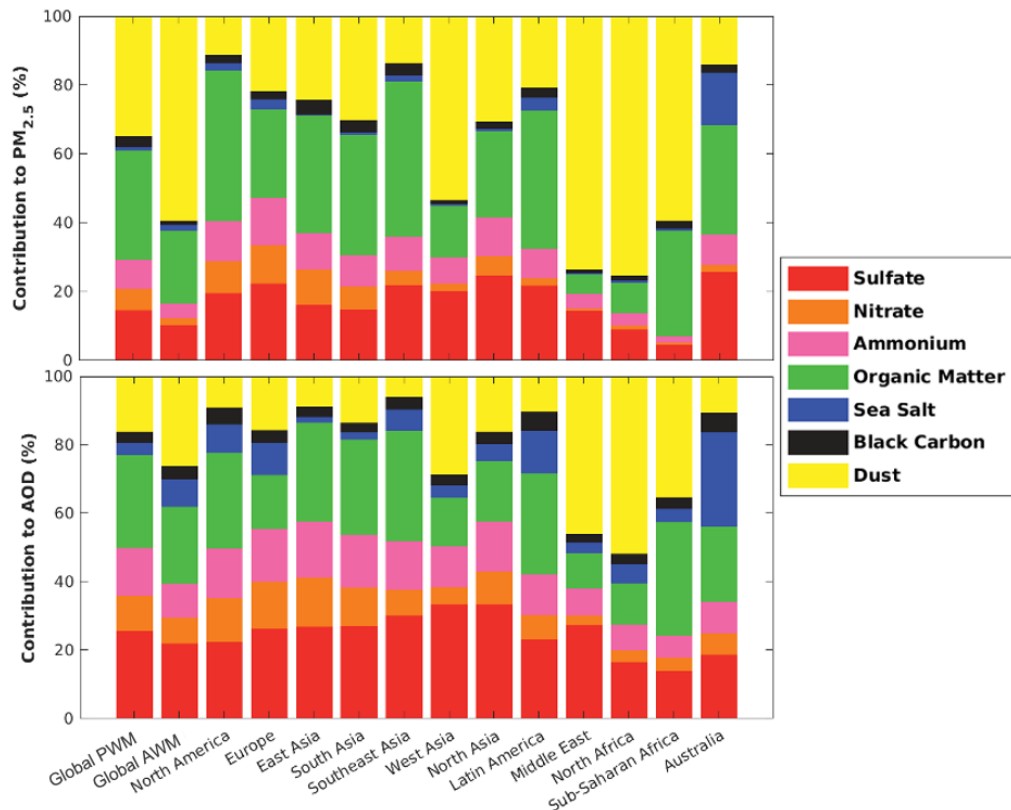


Figure 3. Global and regional PWM contributions of aerosol composition to surface $PM_{2.5}$ (top) and AOD
(bottom). The global area-weighted mean (AWM) over land is also included as the second column.
**3.3    Spatial Variability in Aerosol Vertical Profile**
Figure 4 shows the AOD fraction below 1 km in the GEOS-Chem simulation. Globally, 35.3% of
the PWM AOD is below 1 km. The PWM value is greater than the AWM value since populated
areas tend to have more surface emissions of particles and precursors. Over North America, Europe,
and East Asia, the PWM surface AOD fractions are much higher than the medians and AWM,
indicating high spatial heterogeneity between urban and remote areas. Europe exhibits the highest
variation and the largest discrepancy between PWM and AWM, reflecting the largest spatial
heterogeneity in aerosol vertical profile, driven by influences from regional pollution, marine



aerosols, and transported dust (Zhao et al., 2018). Southeast Asia has the highest surface AOD
fraction and a large variation. Local sources, long-range transported dust, and the influence of
trade winds all contribute to the unique spatial variation in aerosol vertical profile in this region
(Nguyen et al., 2019; Banerjee et al., 2021). Globally, PWM values exhibit less variation than
AWM, indicating moderate variation in aerosol profile across populous areas.

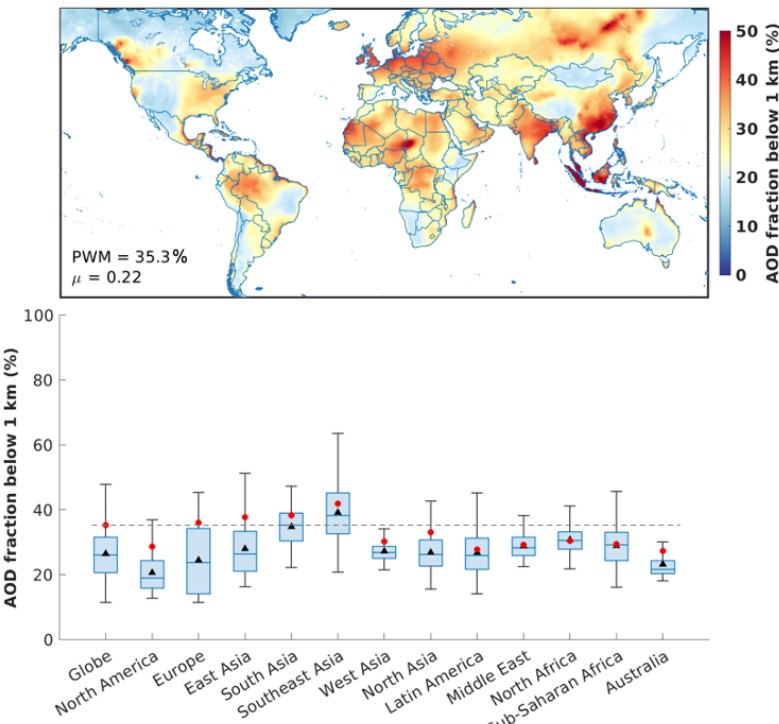


Figure 4. (Top) Map of AOD fraction below 1 km. (Bottom) Global and regional statistics for AOD
fraction below 1 km. Black triangles show the area-weighted mean. Red circles show the PWM. The line
inside each box is the sample median. Each box's top and bottom edges are the 75 and 25 quartiles,
respectively. Vertical bars are the maximum and minimum values within 1.5 times the interquartile range.
The dashed line indicates global PWM.
**3.4    Sensitivity Tests with Globally Uniform Parameters**
Figure 5 shows the global changes in the spatial variation in η due to variations in aerosol chemical
composition (top) and aerosol vertical profile (bottom). Globally, neglect of spatial variation in





aerosol composition induces a 12.3 μg/m$^3$ PWMD in η spatial variation. Both $PM_{2.5}$ and AOD are
strongly affected by aerosol composition, following a similar spatial pattern (Figure A7). Over
mid- and low-latitude areas, the change in AOD is stronger than in $PM_{2.5}$, which gives the opposite
pattern in the η. Neglect of spatial variation in chemical composition reduces η over North Africa
and the Middle East, desert regions where aerosols contain more weakly hygroscopic components
such as mineral dust, compared to populous areas, which contain more secondary inorganic aerosol
(Figure 3). For smaller deserts in the Southwest U.S., Argentina, and Southwest Africa, the dust
fractions of surface aerosols are higher than the global mean (36%, 76%, and 49%, respectively),
but the dust fraction for AOD is similar to the global mean (15%, 25%, and 14%, respectively).
Therefore, neglect of the spatial variation of chemical composition increases η over these small
deserts by increasing the fraction of hygroscopic components in $PM_{2.5}$ and leaving AOD almost
unchanged (Figure A7). It also reduces η over the boreal forests and the Amazon, where surface
aerosols contain little dust and are more hygroscopic compared to populous areas (Figure 3).
Neglect of spatial variation in chemical composition increases η over the eastern U.S. and eastern
China, where $PM_{2.5}$ contains more hygroscopic SNA and less dust than the global mean. It also
increases η in coastal regions where aerosol contains more hygroscopic sea salt than the global
mean.
Neglect of spatial variation in the aerosol vertical profile induces an 8.4 μg/m$^3$ PWMD in η spatial
variation (Figure 5), following the spatial pattern of the change in surface $PM_{2.5}$ (Figure A8). The
most apparent feature is an increase in η throughout the remote northern hemisphere, driven by an
increased aerosol fraction near the surface where the fraction is normally small (Figure 4). The
uniform aerosol vertical profile decreases η over northern Africa and biomass burning regions of
the boreal forests, the Amazon, and Indonesia, driven by a decreased aerosol fraction near the
surface in regions where that fraction is normally high.



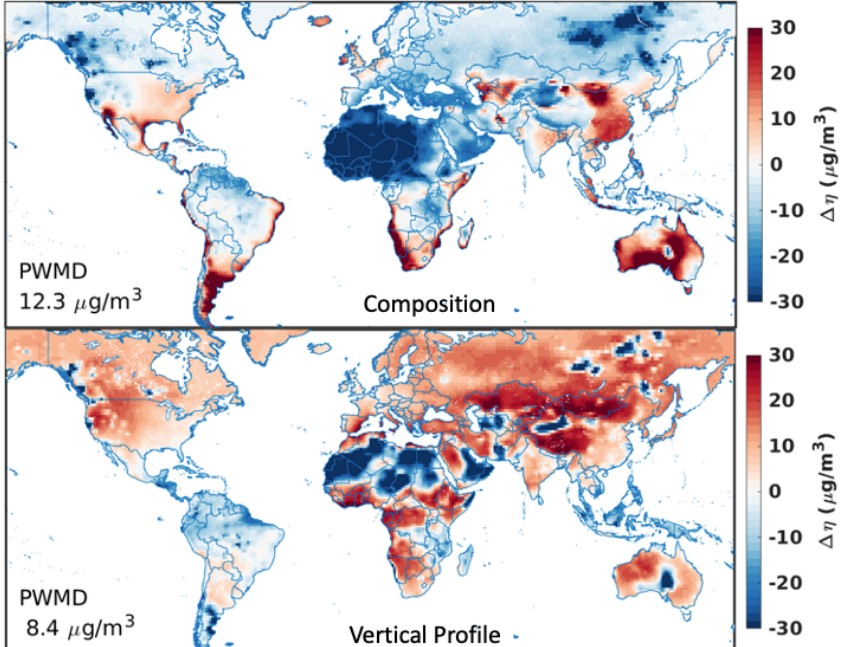


Figure 5. Changes in η (test -base) for each sensitivity test. In the first test, a global PWM aerosol
composition replaces the actual composition (top). In the second test, a global PWM aerosol profile
replaces the actual profiles (bottom). Numbers inset indicate population-weighted mean difference
(PWMD).

## Conclusion

Understanding the global variation of the PM$_{2.5}$ and AOD relationship (η) offers insight into the
geophysical inference of PM$_{2.5}$ from satellite AOD observations. We collected ground-based PM$_{2.5}$
measurements from 6188 sites and MODIS MAIAC satellite AOD throughout the year 2019 to
obtain, for the first time, a global scale observationally based η map. Observed annual mean η
ranges from 7.8 μg/m$^3$ in Hawaii to 321 μg/m$^3$ in Central Asia. We observed enhanced η of 132
μg/m$^3$ to 154 μg/m$^3$ over arid regions such as North Africa and West Asia, due to their low aerosol
extinction efficiency. Moderate η of 97 μg/m$^3$ to 121 μg/m$^3$ was found in industrial areas such as
East Asia and South Asia, where anthropogenetic emissions increase the near-surface PM$_{2.5}$
concentrations. Over remote areas, low η (< 50 μg/m$^3$) was usually observed.



We simulated the global annual mean η with the GEOS-Chem chemical transport model in its high
performance configuration (GCHP). The simulation generally represented observed η with PWM
within 4% (99.5 μg/m$^3$ vs 95.7 μg/m$^3$), a correlation of 0.64 over the 6,118 measurement sites, and
a slope of 0.81. We examined the correlation between simulation and measurements to identify
the two most impactful drivers for η spatial variation - aerosol composition and aerosol vertical
profile, both of which strongly affect the annual mean relation of columnar AOD at ambient RH
with surface PM$_{2.5}$ at controlled RH of 35%. We conducted sensitivity tests by eliminating the
spatial variation of each driver and quantified the impact on η spatial variability. Imposing a
globally uniform aerosol composition led to more pronounced changes (PWMD = 12.3 μg/m$^3$)
reflecting how changes in aerosol composition affect both AOD and surface PM$_{2.5}$, due to the
effects of aerosol hygroscopicity on both quantities. Imposing a globally uniform aerosol vertical
profile had a moderate effect (PWMD = 8.4 μg/m$^3$), reflecting changes in the fraction of aerosol
near the surface.
These findings motivate additional efforts to develop the simulation of aerosol composition and
aerosol vertical profile. Promising avenues include: (1) enhancing global long-term measurements
of PM$_{2.5}$ chemical composition to evaluate and improve simulations, (2) exploiting new and
emerging information about aerosol type from satellite remote sensing (e.g. PACE, MAIA), (3)
advancing simulations at finer spatial resolution to better represent processes affecting aerosol
composition and vertical profile, (4) leveraging aircraft, lidar, and collected AOD-to-PM$_{2.5}$
measurements for constraints on the vertical profile, and (5) exploiting nascent capabilities in
applying satellite remote sensing (e.g. TROPOMI, TEMPO, GEMS) for top-down constraints on
emissions that affect aerosol composition.



# Appendix

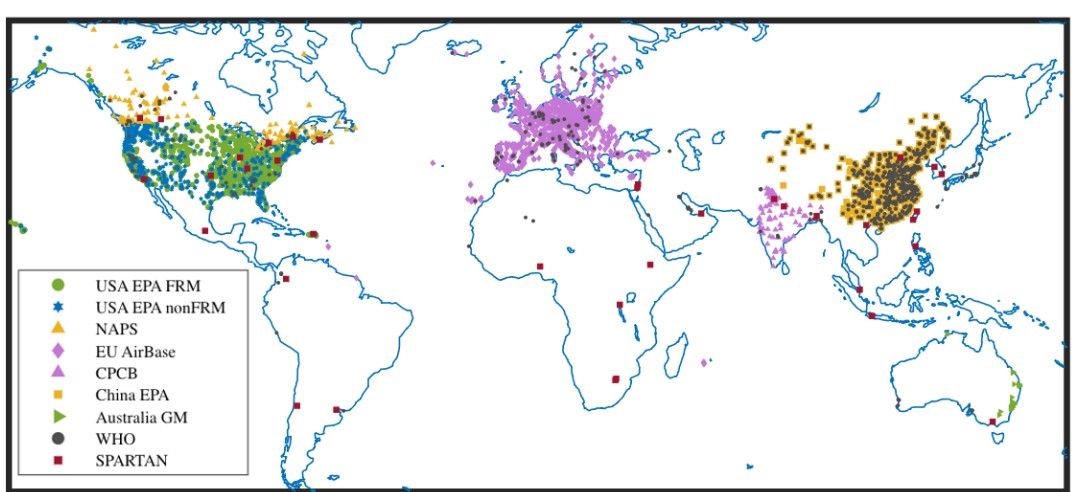

Figure A1. PM$_{2.5}$ measurement sites from publicly available networks.

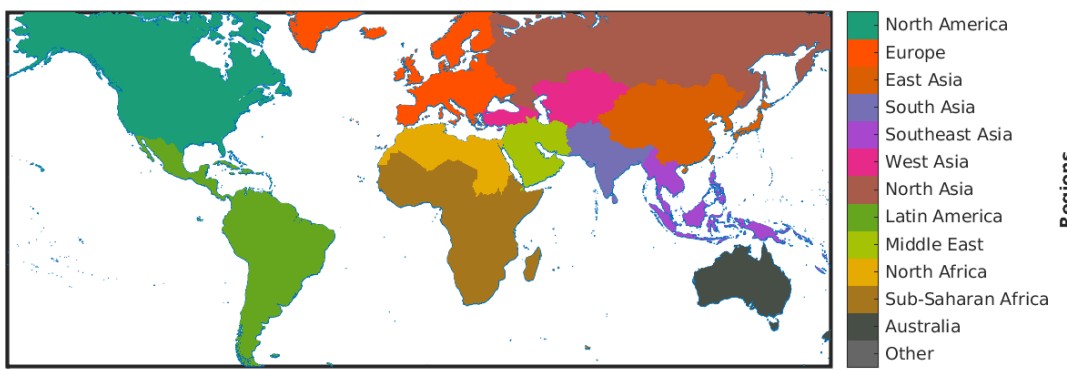

Figure A2. Region definition.

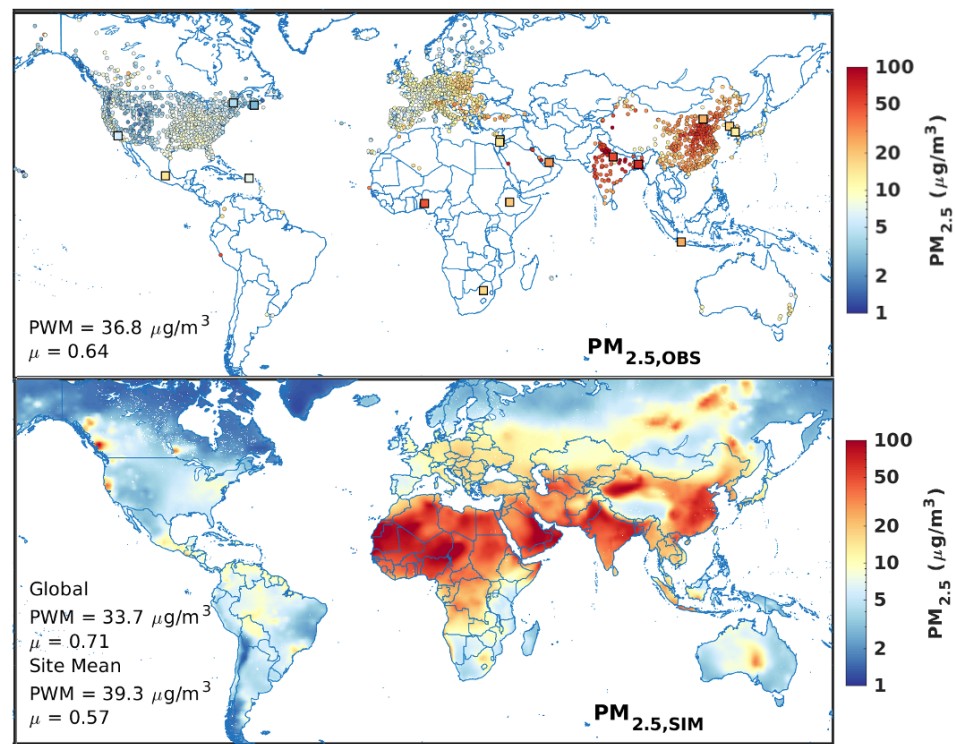

368

Figure A3. Observed (top) and simulated (bottom) annual mean PM$_{2.5}$ for 2019. Circles represent
measurement sites from regional networks or reported by the WHO. Squares represent measured PM$_{2.5}$
from SPARTAN. PWM = population-weighted mean, μ = coefficient of variation.



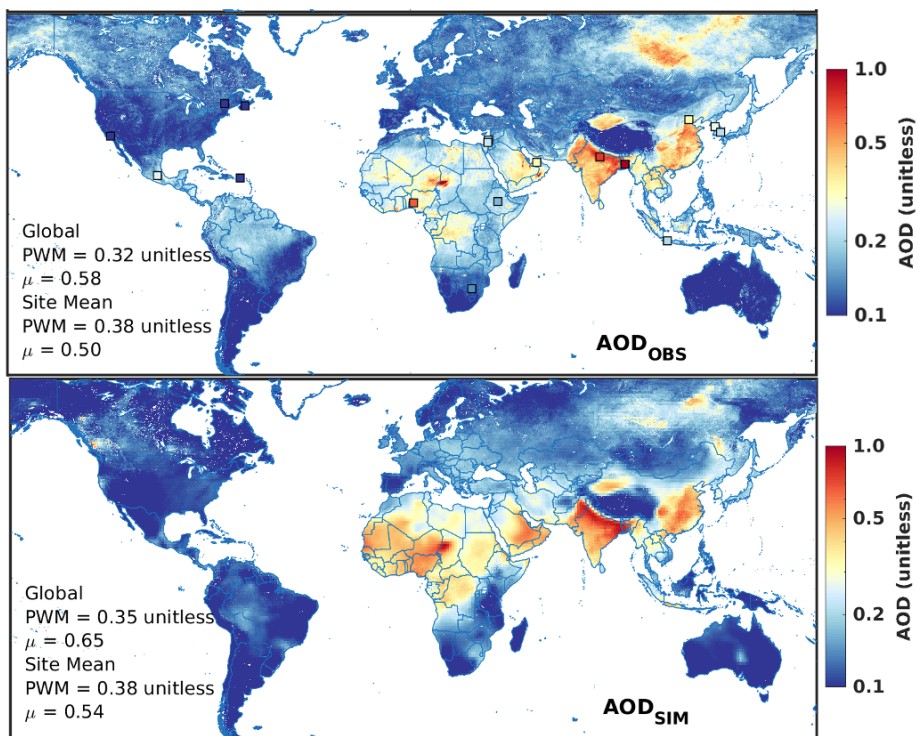

Figure A4. Satellite retrieved (top) and GCHP simulated (bottom) annual mean AOD for 2019. Squares represent ground-measured AOD from AERONET. PWM = population-weighted mean, μ = coefficient of variation.



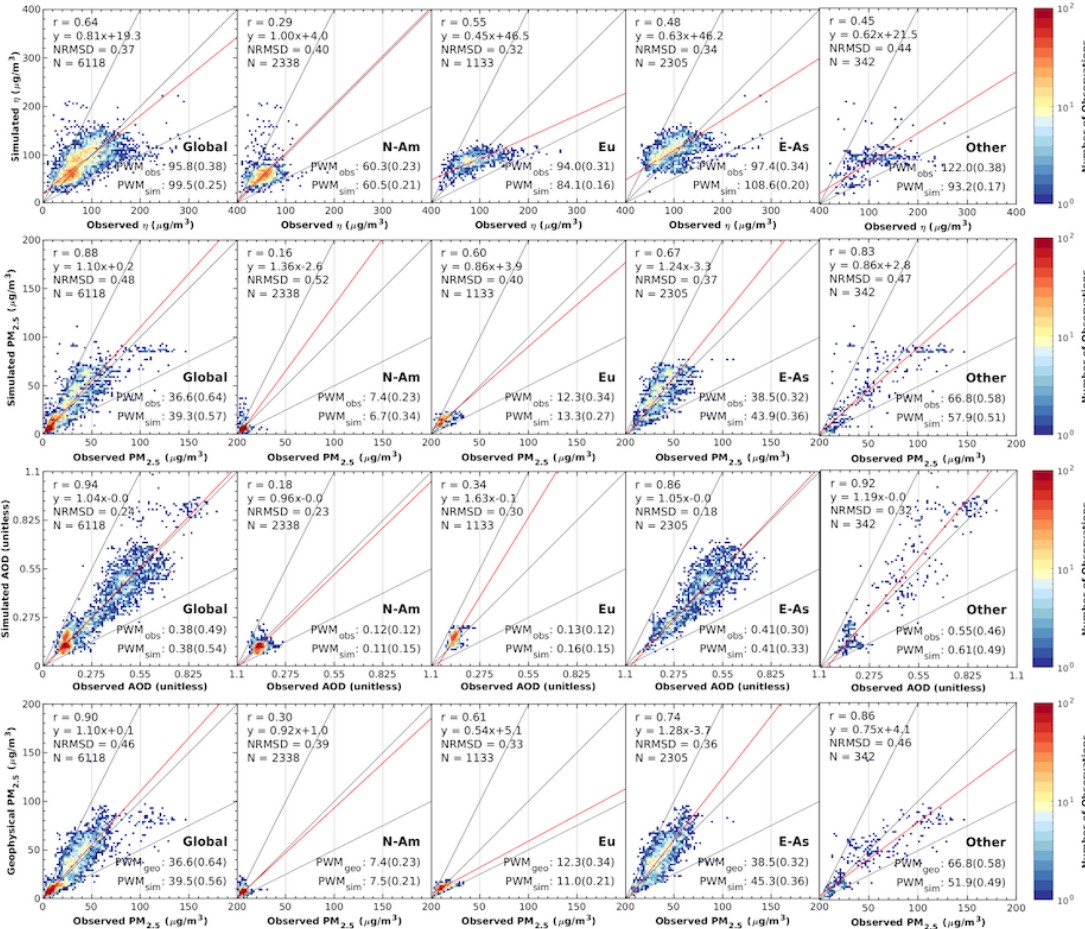


Figure A5. Scatter plots of simulated and observed η (top row), simulated and ground measured $PM_{2.5}$
(second row), simulated and MAIAC AOD (third row), and geophysical and observed $PM_{2.5}$ (bottom
row). The red line shows the line of best fit using Reduced Major Axis Linear Regression. Insets on the
top left show the coefficient of determination ($R^2$), line of best fit, normalized root mean square deviation
(NRMSD), and total number of data points (N). The bottom right insets show the population-weighted
mean of observed, simulated, or geophysical estimation of each dataset, coefficients of variation are
bracketed. Detailed regional mean and coefficients of variation for other regions can be found in Table
A1.





Table A1. Regional population-weighted mean η, PM₂.₅, and AOD from both observation and
simulations. Geophysical PM₂.₅ is also included. Coefficients of variation are bracketed. Regional mean
and coefficients of variation for North America, Europe, and East Asia can be found in Figure A5.

| Region | | South Asia | Southeast Asia | West Asia | Latin America | Middle East | North Africa | Sub-Sahara Africa | Australia |
|---|---|---|---|---|---|---|---|---|---|
| **Number of sites** | | 162 | 3 | 43 | 2 | 46 | 29 | 3 | 5 |
| **η** [μg/m³] | Observed | 121.6 (0.37) | 128.6 (0.12) | 154.0 (0.23) | 72.0 (0.29) | 94.1 (0.56) | 132.3 (0.35) | 196.0 (0.01) | 133.9 (0.34) |
| | Simulated | 93.4 (0.10) | 82.4 (0.09) | 93.4 (0.03) | 74.1 (0.04) | 83.6 (0.21) | 126.6 (0.17) | 105.9 (0.01) | 187.3 (0.26) |
| **PM₂.₅** [μg/m³] | Observed | 81.0 (0.41) | 35.7 (0.44) | 22.0 (0.21) | 12.0 (0.23) | 21.7 (0.51) | 28.7 (0.61) | 24.0 (0.00) | 35.5 (0.29) |
| | Simulated | 70.2 (0.30) | 31.8 (0.20) | 20.8 (0.08) | 20.9 (0.06) | 10.2 (0.25) | 38.3 (0.53) | 16.7 (0.03) | 90.0 (0.31) |
| | Geo-physical | 62.7 (0.30) | 22.7 (0.29) | 13.9 (0.08) | 12.4 (0.08) | 20.4 (0.37) | 27.3 (0.49) | 12.9 (0.03) | 58.1 (0.74) |
| **AOD** [unitless] | Observed | 0.67 (0.25) | 0.27 (0.35) | 0.14 (0.08) | 0.17 (0.03) | 0.24 (0.21) | 0.21 (0.28) | 0.12 (0.01) | 0.30 (0.66) |
| | Simulated | 0.73 (0.28) | 0.38 (0.18) | 0.22 (0.09) | 0.28 (0.02) | 0.12 (0.14) | 0.29 (0.32) | 0.16 (0.01) | 0.51 (0.26) |


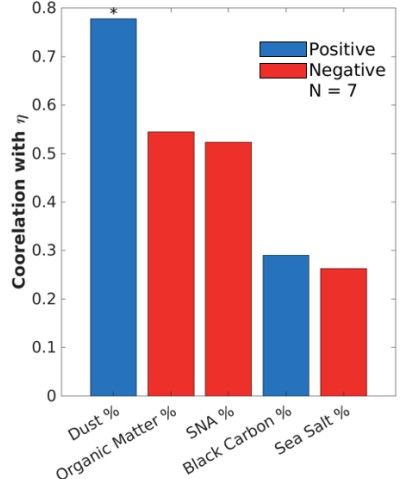




391 Figure A6. Correlation with η of ground-measured aerosol fractional composition from SPARTAN.

392 Organic matter is inferred through residual (Snider et al., 2016). Blue bars indicate positive correlations.

393 Red bars indicate negative correlations. Stars above each bar indicate the p-value associated with each

394 correlation. '***' means the p-value is lower than 0.001, '**' means lower than 0.01, and '*' means

395 lower than 0.5.

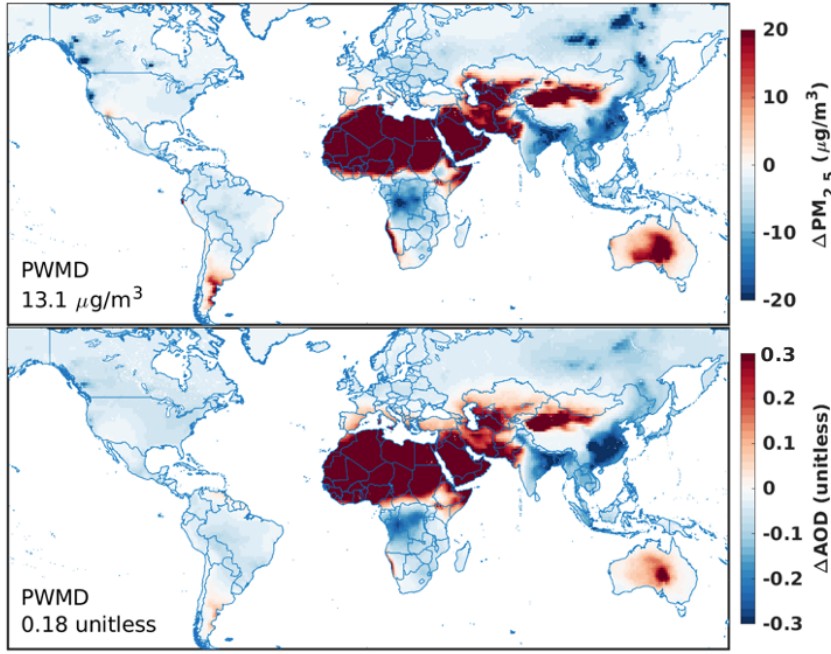

396

397 Figure A7. Changes in $PM_{2.5}$ (top) and AOD (bottom) (test - base) when imposing a global PWM aerosol

398 composition.

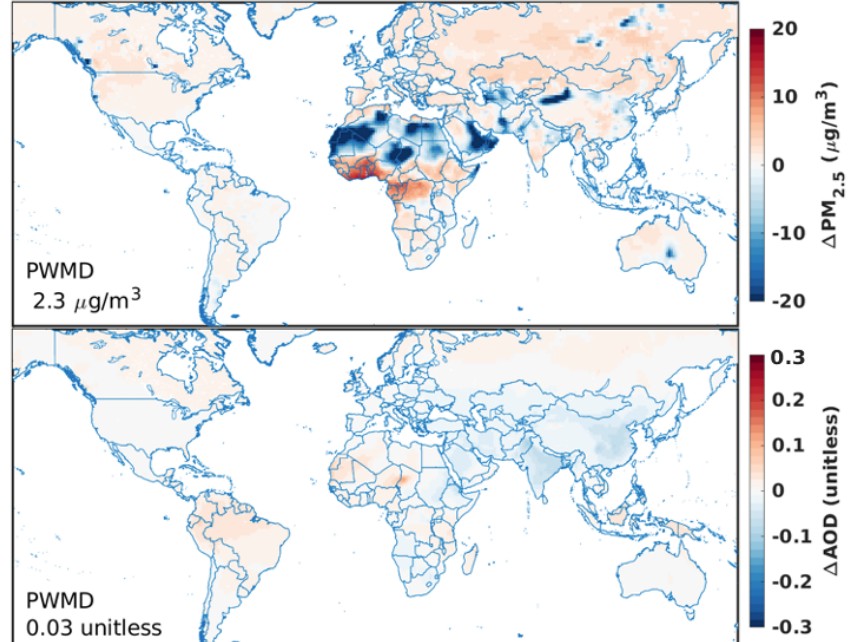

399

Figure A8. Changes in PM$_{2.5}$ (top) and AOD (bottom) (test - base) when imposing a global PWM aerosol profile.

*Data availability.* GEOS-Chem in its high-performance configuration version 13.4.0 can be downloaded at https://zenodo.org/records/6564711.

*Author contributions.* HZ and RVM designed the study. HZ performed the data analysis and model simulation with contributions from AvD, CL, YL, DZ, JM, MH & IS. AvD contributed to the compiled the MAIAC AOD dataset and ground-based observation datasets for PM$_{2.5}$. AL contributed to the original MAIAC AOD dataset. CRO and XL contributed to the SPARTAN data utilization and analysis. The manuscript was written by HZ and RVM with contributions from all authors.

*Competing interests.* The authors declare no competing financial interest.

*Acknowledgment.* This work was supported by NASA Grant 80NSSC22K0200. We thank Dr. Mi Zhou from Princeton University for providing ground based PM$_{2.5}$ data over India.

413



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
