# Peer review of "Importance of Aerosol Composition and Aerosol Vertical Profile in Global Spatial Variation of the PM2.5 to AOD Relationship"

_EGUsphere, 2024_

## Author Comment (AC1)

**Reviewer 1: general overview:**

The manuscript (egusphere-2024-950) entitled "Global Spatial Variation in the PM2.5 to AOD Relationship Strongly Influenced by Aerosol Composition" presented an attractive aspect of the relation between PM2.5 and AOD. Under the consideration of such relationships, as the manuscript title itself declared, PM2.5 composition is quite an important point. However, the current presentation severely lacked the description and validation of them. Without the clarification of these points, I do not go through to further discussion points. Please address the following comments before the possible publication from the journal of ACP.

We appreciate reviewer 1 for the valuable suggestions. We revised the manuscript to incorporate additional datasets, more descriptions of our methods, and further discussion of the results. To address reviewer 1's major concern, an additional figure (Figure A2) comparing GCHP simulated surface PM2.5 chemical composition to ground measurements, is included in the Appendix. Please find our detailed response to each comment below.

**critical concerns:**

From Section 2.4, we can follow the description of PM2.5 components. However, for example, "A 50% reduction of the surface nitrate concentration is applied to account for the long persisting bias in surface nitrate simulated by GEOS-Chem" was quite an unusual treatment considering the modeling results. In addition, "We artificially increase simulated AOD by 0.04 globally to address a poorly understood systematic bias." also seems to be a trick. Despite such unusual post-analysis for PM2.5 components, we can only find the modeling evaluation for  $\eta$ , PM2.5, and AOD. It is desired to present the modeling evaluation for PM2.5 components such as SNA, black carbon, organic matter, sea salt, and dust. Without the detailed information for them, the result and discussion based on Fig. 3 cannot be understood.

To demonstrate the model's performance on global scale PM2.5 composition simulation and inform readers about the potential uncertainties in our analysis, we collected publicly available PM2.5 compositional data from USEPA (306 sites), EU Environmental Protection Agency (37 sites), and the Surface PARTiculate mAtter Network (SPARTAN; 22 sites). We added a paragraph in section 2.1, lines 108-112, to clarify the source and coverage of PM2.5 compositional data: "We also collected PM2.5 compositional data to assess GCHP simulated composition. Long-term PM2.5 compositional data are publicly available from the United States Environmental Protection Agency's Air Quality System, the European Environment Agency Air Quality e-Reporting system, and SPARTAN. A total of 365 sites covering the U.S. (306), Europe (37), and the Global South (22) are included."

Our adjustment of 50% reduction in nitrate significantly mitigates the normalized mean bias (NMB) between simulated and ground measured nitrate, when compared an out-of-box GCHP simulation. Here are our results added to the Appendix: "

Figure A2. Normalized mean bias (NMB) between GCHP simulated PM2.5 chemical composition and ground measurements from CSN, IMPROVE, EBAS, and SPARTAN. The original simulation is the outof-box version of GCHP v13.4.0, the updated simulation includes adjustments such as GFED4.1s emission at daily scale, diel variation and vertical distribution of anthropogenic emissions, and 50% reduction in nitrate concentration."

In response to the concern about nitrate reduction, we want to point out GEOS-Chem has been widely used for high-quality studies on PM2.5 composition at global (McDuffie et al., 2021; Weagle et al., 2018) and regional (Geng et al., 2017; Kim et al., 2015; Philip et al., 2014; Zhai et al., 2021) scales. Even though there is a systematic high bias in nitrate, same as other models such as CMAQ (Shimadera et al., 2014), WRF-Chem (Sha et al., 2019), and EMEP/MSC-W(Prank et al., 2016), the variability of composition can reveal valuable information guiding future research. For example, Geng *et al.* (2017) show a correlation of 0.65 between satellite-derived and measurement PM2.5 nitrate over China, leveraging the nitrate fraction from GEOS-Chem. In another study, Zhai *et al.* (2021) show that GEOS-Chem well captures the vertical variability of all PM2.5 composition during KORUS-AQ campaign (Figure 1 in Zhai *et al.* 2021).

Therefore we revised line 154 to 161 to clarify the bias and its impact on our analysis: "A 50% reduction of the surface nitrate concentration is applied to account for the long-standing bias in surface nitrate simulated by GEOS-Chem (Heald et al., 2012; Miao et al., 2020; Travis et al., 2022; Zhai et al., 2021; Zhang et al., 2012; also Figure A2 in this manuscript) and other models such as CMAQ (Shimadera et al., 2014), WRF-Chem (Sha et al., 2019), and EMEP/MSC-W (Prank et al., 2016). Despite this issue, GEOS-Chem can sufficiently represent the variability of nitrate for applications to studies at global (McDuffie et al., 2021; Weagle et al., 2018) and regional (Geng et al., 2017; Kim et al., 2015; Philip et al., 2014; Zhai et al., 2021) scales."

In section 2.4, lines 167 – 174, we added the rationale behind the minor change in AOD, "Global RH-dependent aerosol optical properties are based on the Global Aerosol Data Set (GADS) (Kopke P., 1997), as originally implemented by Martin et al. (2003), with updates for SNA and OM dry size (Zhu et al., 2023), hygroscopicity (Latimer and Martin, 2019), mineral dust size distribution (Zhang et al., 2013), and absorbing brown carbon (Hammer et al., 2016). These

updates enable GEOS-Chem to capture 74% of the AOD spatial variability when evaluated with AERONET AOD, although a slight low bias (slope = 0.94) persists (Zhu et al., 2023). Low bias in simulated AOD is also reported for other models such as CMAQ (Jin et al., 2019) and WRF-Chem (Benavente et al., 2023). We artificially increase simulated AOD by 0.04 globally to address this poorly understood systematic bias that, although minor, is useful for the representation of  $\eta$ ."

**specific comments:**

• Line 14 and Line 46: Under the context of the abstract, the meaning of  $\eta$  is not clear. In addition, the meaning of  $\eta$  is not clear even in the main text. How about defining  $\eta$  with a mathematical formula? Although a clear definition could be followed from the authors' previous studies, this manuscript should be standalone.

To make the definition clearer, we make the following changes:

In the Abstract, on line 14 - "This study aims to understand the spatial pattern and driving factors of the relationship by examining simulated  $\eta \left(=\frac{PM_{2.5}}{AOD}\right)$ ."

In the Introduction, on line 38 to 40 - "A large community relies upon the spatial distribution of PM2.5 concentrations inferred from satellite AOD and a modelled PM2.5 to AOD relationship for health impact assessment and epidemiological analyses of long-term exposure"

In Methods, on line 76 to 78: "We collect ground-based measurements of PM2.5 for the year 2019 from which to produce observational constraints on  $\eta \left(\frac{PM_{2.5}}{AOD}\right)$ , the spatially and temporally varying ratio between 24-hour surface PM2.5 concentrations and AOD at satellite sampling time."

• Line 74: Taking into consideration the year-to-year variation of dust, why the year 2019 was focused in this study? Were there severe dust events? The motivation for the selection of a target year will be helpful for readers.

The regions of most interest for this study are the populous regions. We consider anthropogenic emissions a more important factor compared to natural dust. The year 2019 was the year with the most ground-based PM2.5 monitoring data and the most recent global-scale anthropogenic emission data when the manuscript was in preparation. Despite the year-to-year variation of natural dust, the relative contribution to PM2.5 in populous regions is less variable.

We add to line 78 to 80 to explain: "At the time of manuscript preparation, the year 2019 offered the greatest density of measurements and the most current emission inventory."

• Line 132-139: The target year is 2019 in this study but did these emission datasets correspond to the year 2019? Or, was there a difference between the simulation year and the emission year? This point should be clarified, and if the latter, how about their impacts on modeling reproducibility?

Yes, the anthropogenic emissions are from CEDSv2 for the year 2019. This was the most recent year available as of March 2024, when the manuscript was submitted. The major natural emissions are fire and dust. We used GFED4 (van der Werf et al., 2017) and offline dust (Meng et al., 2021), both for the year 2019.

To clarify , we revised lines 145 to 146: "The primary emission data are from the Community Emissions Data System version 2 (CEDSV2; Hoesly et al., 2018; CEDS, 2024) for the year 2019." and line 150 to 151 "Biomass burning emissions use the Global Fire Emissions Database, version 4 (GFED4) at daily resolution (van der Werf et al., 2017) for the year 2019."

• Line 171-172: The sentence "P represents population density in each grid box" can be moved into Line 167-168 because these definitions used P.

We revised the manuscript accordingly.

• Line 169-177: The order of i, j, k, and S are confusing in the definitions of F and R. Why the order was different between them? If there is no specific reason, these orders should be unified through variables.

We revised the subscript order to make them consistent, following the order - i, j, k, s.

**technical corrections:**

•Line 255: Correct "coorelation" in this Fig. 2.

Thank you. Typo corrected.

**Reviewer 2:**

Based on analysis of 2019 modeled PM2.5 and AOD, the manuscript claims that global spatial variation in the PM2.5 to AOD relationship is strongly influenced by aerosol composition. The study finds that the relationship is affected by aerosol composition and vertical profile. Although this seems to be an important thing, the manuscript has drawbacks in supporting the conclusions, which must be addressed before publication.

We appreciate reviewer 2 for the valuable comments. The manuscript is revised to incorporate additional  $PM_{2.5}$  ground measurements in the Global South, as well as measured  $PM_{2.5}$  chemical composition from 3 networks. We updated the descriptions of our methods and included further discussion of the results. To address reviewer 2's major concern, an additional figure - Figure A2 - comparing GCHP simulated surface  $PM_{2.5}$  chemical composition to ground measurements, is included in the Appendix. Figure 1, Figure 2, Figure A1, A4 to A6 and Table A1 are updated to reflect to changes. Please find our detailed response to each comment below.

1. Why the ratio of a modeled PM2.5 to AOD is important? Model simulation of PM2.5 is totally based on emissions, meteorological and the modeling algorithm, there is no necessary connection with AOD.

The simulated  $PM_{2.5}$  and AOD relationship provides the opportunity to derive surface  $PM_{2.5}$  using observationally based data (satellite observed total column AOD). The satellite-derived  $PM_{2.5}$  can fill gaps in ground-based monitoring. Understanding and better simulating the  $PM_{2.5}$  and AOD relationship will provide a better estimation of surface  $PM_{2.5}$ , which is particularly important for regions with sparse ground-based  $PM_{2.5}$  measurements. We revised lines 36 to 41 to emphasize the importance of this relationship:

"Satellite remote sensing of aerosol optical depth (AOD), an optical measure of aerosol abundance, offers information about the distribution of  $PM_{2.5}$  (Kondragunta et al., 2022). A large community relies upon the spatial distribution of  $PM_{2.5}$  concentrations inferred from satellite AOD and a modeled  $PM_{2.5}$  to AOD relationship for health impact assessment and epidemiological analyses of long-term exposure (Brauer et al., 2024; Burnett et al., 2018; Cohen et al., 2017; Hao et al., 2023)."

2. GEOS-Chem settings are not detailed enough. What is the resolution? How are the emissions processed? How is the model performance against observations of chemical composition? We have no idea if the model results are good enough for following analysis with only validation of PM total mass.

The model resolution is elaborated in the first paragraph of section 2.4, lines 137-139:

"The simulation is conducted for the year 2019, on a C90 cubed-sphere grid corresponding to a horizontal resolution of about 100 km, with a spin-up time of 1 month."

The emission information, including the anthropogenic ones and the natural ones are elaborated in the second paragraph in section 2.4, from line 143 to 151: "Emissions are processed with the Harmonized Emissions Component (HEMCO) (Lin et al., 2021). The primary emission data are from the Community Emissions Data System version 2 (CEDSV2; Hoesly et al., 2018; CEDS, 2024) for the year 2019. Emissions from stacks are distributed vertically (Bieser et al., 2011). Diel variation of anthropogenic emissions is included (Li et al., 2023). Resolution-dependent soil NOx, sea salt, biogenic VOC, and natural dust emissions are calculated offline at native meteorological resolution to produce consistent emissions across resolutions (Meng et al., 2021; Weng et al., 2020). Biomass burning emissions use the Global Fire Emissions Database, version 4 (GFED4) at daily resolution (van der Werf et al., 2017) for the year 2019."

To demonstrate the model's performance on global scale PM2.5 composition simulation and inform readers about the potential uncertainties in our analysis, we collected publicly available PM2.5 compositional data from USEPA (306 sites), EU Environmental Protection Agency (37 sites), and the Surface PARTiculate mAtter Network (SPARTAN; 22 sites). We added a paragraph in section 2.1, lines108-112, to clarify the source and coverage of PM2.5 compositional data: "We also collected PM2.5 compositional data to assess GCHP simulated composition. Long-term PM2.5 compositional data are publicly available from the United States Environmental Protection Agency's Air Quality System, the European Environment Agency Air Quality e-Reporting system, and SPARTAN. A total of 365 sites covering the U.S. (306), Europe (37), and the Global South (22) are included."

Our adjustment significantly mitigates the normalized mean bias (NMB) between simulated and ground measured nitrate, when compared an out-of-box GCHP simulation. Here are our results added to the Appendix: "

Figure A2. Normalized mean bias (NMB) between GCHP simulated PM2.5 chemical composition and ground measurements from CSN, IMPROVE, EBAS, and SPARTAN. The original simulation is the out-of-box version of GCHP v13.4.0, the updated simulation includes adjustments such as

GFED4.1s emission at daily scale, diel variation and vertical distribution of anthropogenic emissions, and 50% reduction in nitrate concentration."

We therefore added the following to section 3.1, line 237-240: "Evaluation of the simulation of  $PM_{2.5}$  chemical composition versus ground-based measurements reveals a high degree of consistency (Figure A2; normalized mean bias = -0.67 to 0.94) that supports their further assessment of the factors affecting  $\eta$ ."

3. The study does not use all available observations. With very sparse observations sites in Africa, South American, it is quite challenging to obtain solid conclusions as claimed.

We found and added to our analysis 752 more  $PM_{2.5}$  sites. This includes 58 more sites in South Asia, 96 more sites in the Middle East, 3 more sites in Africa, although no new sites in South America. We updated Figure 1, Figure 2, Figure A1, Figure A4-A6, Table A1, and the relevant texts to reflect this update.

4. The importance of chemical composition or vertical profile is simply depending on a sensitivity test to a global uniformed value. There is no logic here. It can't be claimed by just Figure 5 with a sensitivity test and global averages. I believe you can find other things important if you do a sensitivity test.

We clarify that the importance of chemical composition and vertical profile is determined by Figure 2, not Figure 5. Figure 2 examines other factors, such as boundary layer height, wind speed, and surface temperature. Based on these results, we narrow down to aerosol chemical composition and aerosol vertical profile for a more detailed analysis - the sensitivity tests in Figure 5. Although the sensitivity tests can certainly be done with more details (e.g., for certain regions and seasons), our goal is to addresses a global-scale study focusing on annual mean representation. The results are elaborated and explained in lines 254 to 264: "Figure 2 shows the spatial correlation of annual mean factors versus observation-based n. Aerosol components, particularly those with strong primary sources (dust, OM, and black carbon), exhibit the strongest correlations (>0.27) with observationally based n. Significant positive correlations are found for mineral dust and black carbon, both of which are non- or weakly-hygroscopic. Significant negative correlations are found for organic matter and sea salt, reflecting a weak connection between surface concentrations and AOD aloft. Processes are further discussed in sections 3.2 and 3.4. The aerosol vertical profile exhibits a moderate correlation with n (0.14), which is notably higher than any meteorological factors (<0.10). Ground-based data from SPARTAN and AERONET corroborate the correlation between aerosol composition and  $\eta$  (Figure A7). We thus focus further analysis in Sections 3.2-3.4 on the two main drivers in n: aerosol composition and aerosol vertical profile."

We also revised lines 22 to 25 for clarification: "The spatial correlation of observed  $\eta$  with meteorological fields, aerosol vertical profiles, and aerosol chemical composition reveals that the spatial variation of  $\eta$  is strongly influenced by aerosol composition and aerosol vertical profile."

5. Vertical profile also influences the results a lot (8.4 compared to 12.3 of chemical composition), I think it is worth noting in the title.

[revised manuscript text omitted]

---

## Author Comment (AC2)

**The first uploaded version of the Author Comments was incorrect. Please use this version.**

**Reviewer 1:**
**general overview:**

The manuscript (egusphere-2024-950) entitled "Global Spatial Variation in the PM2.5 to AOD Relationship Strongly Influenced by Aerosol Composition" presented an attractive aspect of the relation between PM2.5 and AOD. Under the consideration of such relationships, as the manuscript title itself declared, PM2.5 composition is quite an important point. However, the current presentation severely lacked the description and validation of them. Without the clarification of these points, I do not go through to further discussion points. Please address the following comments before the possible publication from the journal of ACP.

We appreciate the affirmation of the importance of this work and the valuable suggestions. We revised the manuscript to incorporate additional datasets, more descriptions of our methods, and further discussion of the results. To address reviewer 1's major concern, an additional figure (Figure A2) comparing GCHP simulated surface $PM_{2.5}$ chemical composition to ground measurements, is included in the Appendix. Please find our detailed response in blue to each comment below.

**critical concerns:**

From Section 2.4, we can follow the description of PM2.5 components. However, for example, "A 50% reduction of the surface nitrate concentration is applied to account for the long persisting bias in surface nitrate simulated by GEOS-Chem" was quite an unusual treatment considering the modeling results. In addition, "We artificially increase simulated AOD by 0.04 globally to address a poorly understood systematic bias." also seems to be a trick. Despite such unusual post-analysis for PM2.5 components, we can only find the modeling evaluation for η, PM2.5, and AOD. It is desired to present the modeling evaluation for PM2.5 components such as SNA, black carbon, organic matter, sea salt, and dust. Without the detailed information for them, the result and discussion based on Fig. 3 cannot be understood.

To evaluate the model's performance in representing global scale $PM_{2.5}$ composition simulation, we collected publicly available $PM_{2.5}$ compositional data from USEPA (306 sites), EU Environmental Protection Agency (37 sites), and the Surface PARTiculate mAtter Network (SPARTAN; 22 sites). We added a paragraph in section 2.1, lines 108-112, to clarify the source and coverage of $PM_{2.5}$ compositional data: "We also collected publicly available $PM_{2.5}$ compositional data to assess GCHP simulated composition. Long-term $PM_{2.5}$ compositional data are included from the United States Environmental Protection Agency's Air Quality System, the European Environment Agency Air Quality e-Reporting system, and SPARTAN, with a total of 365 sites covering the U.S. (306), Europe (37), and the Global South (22)."

Our adjustment of 50% reduction in nitrate significantly mitigates the normalized mean bias (NMB) between simulated and ground measured nitrate, when compared an out-of-box simulation. Here are our results added to the Appendix: "

[Figure]

Figure A2. Normalized mean bias (NMB) between simulated PM$_{2.5}$ chemical composition and ground measurements from CSN, IMPROVE, EBAS, and SPARTAN. The original simulation is the out-of-box version of GCHP v13.4.0, the updated simulation includes adjustments such as GFED4.1s emission at daily scale, diel variation and vertical distribution of anthropogenic emissions, and 50% reduction in nitrate concentration."

In response to the concern about nitrate reduction, we note that GEOS-Chem has been widely used for high-quality studies on PM$_{2.5}$ composition at global (McDuffie et al., 2021; Weagle et al., 2018) and regional (Geng et al., 2017; Kim et al., 2015; Philip et al., 2014; Zhai et al., 2021) scales. Even though there is a systematic high bias in nitrate, as in other models such as CMAQ (Shimadera et al., 2014), WRF-Chem (Sha et al., 2019), and EMEP/MSC-W(Prank et al., 2016), the simulations exhibit skill in representing nitrate. For example, Geng *et al.* (2017) found that the nitrate fraction from GEOS-Chem was useful to produce a nitrate PM$_{2.5}$ product. In another study, Zhai *et al.* (2021) show that GEOS-Chem well captures the vertical variability of all PM$_{2.5}$ composition during KORUS-AQ campaign (Figure 1 in Zhai *et al.* 2021).

Therefore we revised line 154 to 161 to clarify the bias and its impact on our analysis: "A 50% reduction of the surface nitrate concentration is applied to account for the long-standing bias in surface nitrate simulated by GEOS-Chem (Heald et al., 2012; Miao et al., 2020; Travis et al., 2022; Zhai et al., 2021; Zhang et al., 2012; also Figure A2 in this manuscript) and other models such as CMAQ (Shimadera et al., 2014), WRF-Chem (Sha et al., 2019), and EMEP/MSC-W (Prank et al., 2016). Despite this bias, GEOS-Chem can sufficiently represent the variability of nitrate for applications to studies at global (McDuffie et al., 2021; Weagle et al., 2018) and regional (Geng et al., 2017; Kim et al., 2015; Philip et al., 2014a; Zhai et al., 2021) scales."

In section 2.4, lines 167 – 174, we added the rationale behind the minor change in AOD, "These updates enable GEOS-Chem to capture 74% of the AOD spatial variability versus AERONET (Zhu et al., 2023). A slight systematic low bias against MAIAC AOD is found, with an intercept of -0.05 and a population-weighted mean difference (PWMD) of -0.04. Low bias in simulated AOD is also reported for other models, such as CMAQ (Jin et al., 2019) and WRF-Chem (Benavente et al., 2023). We artificially increase simulated AOD by 0.04 globally to address this

poorly understood systematic bias that, although minor, is useful for the representation of η (PWMD reduced from 20.6 μg/m$^3$ to 1.9 μg/m$^3$)".

**specific comments:**

- Line 14 and Line 46: Under the context of the abstract, the meaning of η is not clear. In addition, the meaning of η is not clear even in the main text. How about defining η with a mathematical formula? Although a clear definition could be followed from the authors' previous studies, this manuscript should be standalone.

  To make the definition clearer, we make the following changes:

  In the Abstract, on line 14 - "This study aims to understand the spatial pattern and driving factors of the relationship by examining simulated η ($= \frac{PM_{2.5}}{AOD}$)."

  In the Introduction, on line 38 to 40 – "A large community relies upon the spatial distribution of $PM_{2.5}$ concentrations inferred from satellite AOD and a modelled $PM_{2.5}$ to AOD relationship for health impact assessment and epidemiological analyses of long-term exposure"

  In Methods, on line 76 to 78: "We collect ground-based measurements of $PM_{2.5}$ for the year 2019 from which to produce observational constraints on η ($\frac{PM_{2.5}}{AOD}$), the spatially and temporally varying ratio between 24-hour surface $PM_{2.5}$ concentrations and AOD at satellite sampling time."

- Line 74: Taking into consideration the year-to-year variation of dust, why the year 2019 was focused in this study? Were there severe dust events? The motivation for the selection of a target year will be helpful for readers.

  The regions of most interest for this study are the populous regions. We consider anthropogenic emissions a more important factor compared to natural dust. When the manuscript was in preparation, the year 2019 was the year with the most ground-based $PM_{2.5}$ monitoring data and the most recent global-scale anthropogenic emission data. Despite the year-to-year variation of natural dust, the relative contribution to $PM_{2.5}$ in populous regions is less variable.

  We add to line 78 to 80 to explain: "At the time of manuscript preparation, the year 2019 offered the greatest density of measurements and the most current emission inventory."

- Line 132-139: The target year is 2019 in this study but did these emission datasets correspond to the year 2019? Or, was there a difference between the simulation year and the emission year? This point should be clarified, and if the latter, how about their impacts on modeling reproducibility?

Yes, the anthropogenic emissions are from CEDSv2 for the year 2019. This was the most recent year available as of March 2024, when the manuscript was submitted. The major natural emissions are fire and dust. We used GFED4 (van der Werf et al., 2017) and offline dust (Meng et al., 2021), both for the year 2019.

To clarify , we revised lines 144 to 146: "The primary emission data are from the Community Emissions Data System version 2 (CEDS$_{V2}$; Hoesly et al., 2018; CEDS, 2024) for the year 2019." and line 150 to 151 "Biomass burning emissions use the Global Fire Emissions Database, version 4 (GFED4) at daily resolution (van der Werf et al., 2017) for the year 2019."

- Line 171-172: The sentence "P represents population density in each grid box" can be moved into Line 167-168 because these definitions used P.

  We revised the manuscript accordingly.

- Line 169-177: The order of i, j, k, and S are confusing in the definitions of F and R. Why the order was different between them? If there is no specific reason, these orders should be unified through variables.

  We revised the subscript order to make them consistent, following the order - i, j, k, s.

**technical corrections:**

- Line 255: Correct "coorelation" in this Fig. 2.

  Thank you. Typo corrected.

**Reviewer 2:**

Based on analysis of 2019 modeled PM2.5 and AOD, the manuscript claims that global spatial variation in the PM2.5 to AOD relationship is strongly influenced by aerosol composition. The study finds that the relationship is affected by aerosol composition and vertical profile. Although this seems to be an important thing, the manuscript has drawbacks in supporting the conclusions, which must be addressed before publication.

We appreciate the affirmation of the importance of this work and the valuable comments. The manuscript is revised to incorporate additional PM$_{2.5}$ ground measurements in the Global South, as well as measured PM$_{2.5}$ chemical composition from 3 networks. We updated the descriptions of our methods and included further discussion of the results. To address reviewer 2's major concern, an additional figure - Figure A2 - comparing GCHP simulated surface PM$_{2.5}$ chemical composition to ground measurements, is included in the Appendix. Figure 1, Figure 2, Figure A1, A4 to A6 and Table A1 are updated to reflect addition of PM$_{2.5}$ ground measurements. Please find our detailed response to each comment below.

1.  Why the ratio of a modeled PM2.5 to AOD is important? Model simulation of PM2.5 is totally based on emissions, meteorological and the modeling algorithm, there is no necessary connection with AOD.

    The simulated PM$_{2.5}$ and AOD relationship provides the opportunity to derive surface PM$_{2.5}$ using observationally based data (satellite observed total column AOD). The satellite-derived PM$_{2.5}$ can fill gaps in ground-based monitoring. Understanding and better simulating the PM$_{2.5}$ and AOD relationship will provide a better estimation of surface PM$_{2.5}$, which is particularly important for regions with sparse ground-based PM$_{2.5}$ measurements. We revised lines 36 to 41 to emphasize the importance of this relationship:

    "Satellite remote sensing of aerosol optical depth (AOD), an optical measure of aerosol abundance, offers information about the distribution of PM$_{2.5}$ (Kondragunta et al., 2022). A large community relies upon the spatial distribution of PM$_{2.5}$ concentrations inferred from satellite AOD and a modeled PM$_{2.5}$ to AOD relationship for health impact assessment and epidemiological analyses of long-term exposure (Brauer et al., 2024; Burnett et al., 2018; Cohen et al., 2017; Hao et al., 2023)."

2.  GEOS-Chem settings are not detailed enough. What is the resolution? How are the emissions processed? How is the model performance against observations of chemical composition? We have no idea if the model results are good enough for following analysis with only validation of PM total mass.

    The model resolution is elaborated in the first paragraph of section 2.4, lines 137-139:

    "The simulation is conducted for the year 2019, on a C90 cubed-sphere grid corresponding to a horizontal resolution of about 100 km, with a spin-up time of 1 month."

The emission information, including anthropogenic and natural are elaborated in the second paragraph in section 2.4, from line 143 to 151: "Emissions are processed with the Harmonized Emissions Component (HEMCO) (Lin et al., 2021). The primary emission data are from the Community Emissions Data System version 2 (CEDS$_{V2}$; Hoesly et al., 2018; CEDS, 2024) for the year 2019. Emissions from stacks are distributed vertically (Bieser et al., 2011). Diel variation of anthropogenic emissions is included (Li et al., 2023). Resolution-dependent soil NO$_x$, sea salt, biogenic VOC, and natural dust emissions are calculated offline at native meteorological resolution to produce consistent emissions across resolutions (Meng et al., 2021; Weng et al., 2020). Biomass burning emissions use the Global Fire Emissions Database, version 4 (GFED4) at daily resolution (van der Werf et al., 2017) for the year 2019."

To evaluate the model's performance in representing global scale PM$_{2.5}$ composition simulation and inform readers about the potential uncertainties in our analysis, we collected publicly available PM$_{2.5}$ compositional data from USEPA (306 sites), EU Environmental Protection Agency (37 sites), and the Surface PARTiculate mAtter Network (SPARTAN; 22 sites). We added a paragraph in section 2.1, lines108-112, to clarify the source and coverage of PM$_{2.5}$ compositional data: "We also collected publicly available PM$_{2.5}$ compositional data to assess GCHP simulated composition. Long-term PM$_{2.5}$ compositional data are included from the United States Environmental Protection Agency's Air Quality System, the European Environment Agency Air Quality e-Reporting system, and SPARTAN, with a total of 365 sites covering the U.S. (306), Europe (37), and the Global South (22)."

Our adjustment of 50% reduction in nitrate significantly mitigates the normalized mean bias (NMB) between simulated and ground measured nitrate, when compared an out-of-box simulation. Here are our results added to the Appendix: "

[Figure]

Figure A2. Normalized mean bias (NMB) between simulated PM$_{2.5}$ chemical composition and ground measurements from CSN, IMPROVE, EBAS, and SPARTAN. The original simulation is the out-of-box version of GCHP v13.4.0, the updated simulation includes adjustments such as GFED4.1s emission at daily scale, diel variation and vertical distribution of anthropogenic emissions, and 50% reduction in nitrate concentration."

We therefore added the following to section 3.1, line 238-240: "Evaluation of the simulation of PM$_{2.5}$ chemical composition versus ground-based measurements reveals a high degree of consistency (Figure A2) that supports their further assessment of the factors affecting η."

3. The study does not use all available observations. With very sparse observations sites in Africa, South American, it is quite challenging to obtain solid conclusions as claimed.

   We found and added to our analysis 752 more PM$_{2.5}$ sites. This includes 58 more sites in South Asia, 96 more sites in the Middle East, and 3 more sites in Africa. We updated Figure 1, Figure 2, Figure A1, Figure A4-A6, Table A1, and the relevant texts to reflect this update.

4. The importance of chemical composition or vertical profile is simply depending on a sensitivity test to a global uniformed value. There is no logic here. It can't be claimed by just Figure 5 with a sensitivity test and global averages. I believe you can find other things important if you do a sensitivity test.

   We clarify that the importance of chemical composition and vertical profile is determined by Figure 2, not Figure 5. Figure 2 examines other factors, such as boundary layer height, wind speed, and surface temperature. Based on these results, we narrow down to aerosol chemical composition and aerosol vertical profile for a more detailed analysis – the sensitivity tests in Figure 5. Although the sensitivity tests can certainly be done with more details (e.g., for certain regions and seasons), our goal is to addresses a global-scale study focusing on annual mean representation. The results are elaborated and explained in lines 254 to 264: "Figure 2 shows the spatial correlation of annual mean factors versus observation-based η. Aerosol components, particularly those with strong primary sources (dust, OM, and black carbon), exhibit the strongest correlations (>0.27) with observationally based η. Significant positive correlations are found for mineral dust and black carbon, both of which are non- or weakly-hygroscopic. Significant negative correlations are found for organic matter and sea salt, reflecting a weak connection between surface concentrations and AOD aloft. Processes are further discussed in sections 3.2 and 3.4. The aerosol vertical profile exhibits a moderate correlation with η (0.14), which is notably higher than any meteorological factors (⩽0.10). Ground-based data from SPARTAN and AERONET corroborate the correlation between aerosol composition and η (Figure A7). We thus focus further analysis in Sections 3.2-3.4 on the two main drivers in η: aerosol composition and aerosol vertical profile."

   We also revised lines 22 to 25 for clarification: "The spatial correlation of observed η with meteorological fields, aerosol vertical profiles, and aerosol chemical composition reveals that the spatial variation of η is strongly influenced by aerosol composition and aerosol vertical profile."

   The connection between Figure 5 and Figure 2 is noted again in lines 322-323: "Figure 5 shows the global changes in the spatial variation in η due to variations in aerosol

chemical composition (top) and aerosol vertical profile (bottom), the two main drivers found in Figure 2."

5. Vertical profile also influences the results a lot (8.4 compared to 12.3 of chemical composition), I think it is worth noting in the title.

[revised manuscript text omitted]